



# Quantifying the diurnal variation of atmospheric NO₂ from observations of the Geostationary Environment Monitoring Spectrometer (GEMS)

David P. Edwards[1], Sara Martínez-Alonso[1], Duseong S. Jo[1,2], Ivan Ortega[1], Louisa K. Emmons[1], John J. Orlando[1], Helen M. Worden[1], Jhoon Kim[3], Hanlim Lee[4], Junsung Park[4,5], and Hyunkee Hong[6]

[1]Atmospheric Chemistry Observations & Modeling Laboratory, National Center for Atmospheric Research, Boulder, CO, USA.
[2]Seoul National University, Seoul, South Korea.
[3]Department of Atmospheric Science, Yonsei University, Seoul, South Korea.
[4]Division of Earth Environmental System Science, Pukyong National University, Busan, South Korea.
[5]Smithsonian Astrophysical Observatory, Cambridge, MA, USA.
[6]National Institute of Environmental Research, Seoul, South Korea

*Correspondence to*: David P. Edwards (edwards@ucar.edu)

**Abstract.** The Geostationary Environment Monitoring Spectrometer (GEMS) over Asia is the first geostationary Earth orbit instrument in the virtual constellation of sensors for atmospheric chemistry and composition air quality research and
applications. For the first time, the hourly observations enable studies of diurnal variation of several important trace gas and aerosol pollutants including nitrogen dioxide (NO₂) which is the focus of this work. NO₂ is a regulated pollutant and an indicator of anthropogenic emissions in addition to being involved in tropospheric ozone chemistry and particulate matter formation. We present new quantitative measures of NO₂ tropospheric column diurnal variation which can be greater than 50% of the column amount especially in polluted environments. The NO₂ distribution is seen to change hourly and can be
quite different from what would be seen by a once-a-day low Earth orbit satellite observation. We use GEMS data in combination with TROPOMI satellite and Pandora ground-based remote sensing measurements and MUSICAv0 3D chemical transport model analysis to examine the NO₂ diurnal variation in January and June 2023 over Northeast Asia and Seoul, South Korea, study regions to distinguish the different emissions, chemistry, and meteorological processes that drive the variation. Understanding the relative importance of these processes will be important for including pollutant diurnal
variation in models aimed at determining true pollutant exposure levels for air quality studies. The work presented here also provides a path for investigating similar NO₂ diurnal cycles in the new TEMPO data over North America, and later over Europe with S-4.

## 1 Introduction

Predicting atmospheric air quality (AQ) requires understanding the processes that emit air pollutants, how these are
transported in the atmosphere, the chemical and physical transformations that take place, and the potential impact on health and environment. Satellite observations provide valuable information on these processes, but until recently, measurements



from an individual platform were limited to twice daily at best when relying on observations from low Earth orbit (LEO). This is now changing with daylight hourly observations of atmospheric trace gases and aerosols from the new geostationary Earth orbit (GEO) satellite sensors. The South Korean GEO-KOMPSAT-2/GEMS (Geostationary Korea Multi-Purpose Satellite-2/Geostationary Environment Monitoring Spectrometer) instrument (Kim et al., 2020) has been operational over Asia since February 2020, NASA's EVI-1 TEMPO (Earth Venture Instrument-1 Tropospheric Emissions: Monitoring Pollution) (Zoogman et al., 2017), was launched in April 2023 to monitor North America, and Europe will be covered by ESA/EUMETSAT Sentinel-4 (Bazalguette Courrèges-Lacoste et al., 2013) in 2025. Common objectives for these missions will provide column products for ozone ($O_3$), nitrogen dioxide ($NO_2$), sulfur dioxide ($SO_2$), formaldehyde ($HCHO$), and aerosol optical depth, among others, several times per day at 5–10 km/pixel spatial scales. Together with LEO sensors such as JPSS/CrIS (Joint Polar Satellite System/Cross-track Infrared Sounder) (Han et al., 2013), MetOp/IASI (Infrared Atmospheric Sounding Interferometer) (Clerbaux et al., 2009), and Sentinel 5-P/TROPOMI (TROPOspheric Monitoring Instrument) (Veefkind et al., 2012), the new GEO missions will form an atmospheric composition satellite virtual constellation with nearly continuous Northern Hemisphere coverage and unprecedented capability to meet the needs of AQ research and applications (CEOS, 2019).

The measurement hourly time resolution is the novel perspective provided by the new GEO platforms that enables: (1) investigations of the diurnal processes determining atmospheric composition; (2) improvements in retrieval sensitivity gained with possible longer measurement acquisition dwell times; and (3) the increased probability of obtaining at least some daily cloud-free observations at any given location (e.g., Fishman et al, 2012; Zoogman et al., 2017; Kim et al., 2020). This work investigates the processes that drive the diurnal variation of $NO_2$ over Northeast Asia, and particularly over Seoul, South Korea, using GEMS data in combination with other satellite and ground-based remote sensing measurements and 3D atmospheric chemical transport model (CTM) analysis.

Of the trace gas products that are routinely retrieved from satellite sensor shortwave spectral measurements, $NO_2$ is one of the most reliable due to the relatively strong signal. It plays a central role in atmospheric chemistry and tropospheric $O_3$ and aerosol formation and is photochemically linked with nitrogen oxide (NO) as reactive nitrogen ($NO_x \equiv NO+NO_2$) (Brasseur et al., 1999). $NO_x$ emissions occur primarily as NO and have anthropogenic sources associated with high temperature combustion processes in the power, industry, and transport sectors (e.g., Goldberg et al., 2021; de Foy et al., 2022). Natural sources include lightning, biomass burning, and soil emissions amongst others (e.g., Griffin et al., 2021; Huber et al., 2020). The $NO_2$ minutes-to-hours daytime lifetime in the planetary boundary layer (PBL) also means that it does not become evenly mixed in the atmosphere, and polluted regions, especially urban areas, often show satellite-derived $NO_2$ column enhancements of many times the background level. These products are therefore particularly useful for understanding $NO_x$ and other emissions and their subsequent chemical and physical transformations along with attributing pollution trends over time due to emission regulations and other factors such as the COVID pandemic lockdowns and economic downturns (e.g., Duncan et al., 2013; Levelt et al., 2022; deRuyter de Wildt et al., 2012). This has resulted in a wealth of peer-reviewed



literature based on LEO satellite observations detailing pollutant research and AQ applications and management (e.g., Curier et al., 2014; Duncan et al., 2016; Liu et al, 2017). The partitioning of $NO_x$ between NO and $NO_2$ is in photochemical steady state that establishes on a timescale of minutes during daylight hours. Details are given in Brasseur et al. (1999) and the $NO_x$ ratio can be represented as:

$$[NO] / [NO_2] = j_{NO2} / ( k_{O3} [O_3] + k_{HO2} [HO_2] + k_{RO2} [RO_2] )$$

where the square bracket denotes concentration (molec.cm$^{-3}$), k is a bimolecular rate coefficient (cm$^3$.molec$^{-1}$.sec$^{-1}$), and j is the photolysis rate (sec$^{-1}$). $HO_2$ is the hydroperoxy radical and $RO_2$ represents all organic peroxy radicals. The main loss of $NO_x$ is through oxidation to the nitrogen reservoirs nitric acid ($HNO_3$) and peroxyacetyl nitrate (PAN). The chemical diurnal cycle is discussed further in Section 5.3 based on the GEMS $NO_2$ data and modeling.

    Following this Introduction, Section 2 describes the remote sensing measurements and the model tools that are used in this
work. Section 3 presents the $NO_2$ diurnal variation observed by GEMS at different spatial scales for our January and June 2023 study months, and this is compared with ground-based remote sensing measurements over Seoul in Section 4. Model analysis and discussion in Section 5 considers the various processes (emissions, chemistry, and meteorology) that drive the $NO_2$ diurnal variation. Conclusions are presented in Section 6.

**2 Observations and modelling tools**

**2.1 GEMS**

    South Korea's GEMS is the first satellite instrument in the GEO constellation and is monitoring AQ over Asia. GEMS was launched in February 2020 by Arianespace from the French Guiana Space Center. Like the TEMPO instrument, GEMS was built by Ball Aerospace & Technologies Corp. GEMS retrievals of $O_3$, $NO_2$, $SO_2$, HCHO, glyoxal (CHOCHO), and aerosols are derived from ultraviolet-visible (UV-vis) measurements (Kim et al., 2020), and the cloud fraction necessary for data
filtering is also available for each observation (Choi et al., 2020; Kim et al., 2024). Each day, the field of regard (FOR) shifts westward with the Sun providing measurements over India at the end of the day at the expense of losing coverage over Japan when the solar zenith angles become too large. Total column (TotC), stratospheric column (StrC), and for some products, tropospheric column (TrC) values are retrieved in up to 10 hourly observations during daytime according to the season with spatial resolution at Seoul near 7 x 7.7 km$^2$ for gases and cloud, 3.5 x 7.7 km$^2$ for aerosol and surface reflectivity.

The main retrieval challenges for $NO_2$ (e.g., Palmer et al., 2001; Buscela et al., 2013; Lorente et al., 2017; Geddes et al., 2018; Zara et al., 2018; van Geffen et al., 2020, Park, 2022b) are the conversion of the observed $NO_2$ slant column density to an inferred vertical column density using an air mass factor (AMF), and the separation of the StrC and TrC. These steps can both lead to significant uncertainties and biases as is well-documented in studies comparing the varying results from different $NO_2$ retrieval algorithms using the same OMI (Ozone Monitoring Instrument, Levelt et al., 2018) satellite



measurements (Zara et al., 2018). The AMF is primarily a geometric conversion based on the observation angles, but this must also consider other factors including cloud and aerosol information, terrain reflectivity, and vertical gas profile (Lorente et al., 2017). A CTM is used in this AMF calculation and also in the separation of the StrC and TrC (Lee et al., 2020; Geddes et al., 2018). Although beyond the scope of this paper, retrieval sensitivity studies and product validation for the new GEO composition measurements will be important to minimize any aliasing of diurnal variation in the retrieval input parameters

onto the diurnal variation of the retrieved products themselves (e.g. Yang et al., 2023a; Kim et al., 2023; Szykman and Liu, 2023). This includes, for example, parameters that impact the AMF, a priori that changes by location and time, angular dependences of surface and cloud reflectivities, vertical profiles of trace gases and aerosols, meteorology, and PBL evolution.

This work uses the publicly available Version 2 (V2.0) NO$_2$ Level 2 data obtainable from the Korean National Institute of

Environmental Research (NIER) (NIER, 2024). An example coverage is shown in Fig. 1. The data quality flags used are those recommended by the algorithm development team (Lee et al., 2022). These are: FinalQualityFlag ≤1; CloudFraction <0.3; and SolarZenithAngle and ViewingZenithAngle ≤70°. Updates and enhancements to the GEMS retrieval algorithms are ongoing and will result in new data versions being released periodically.

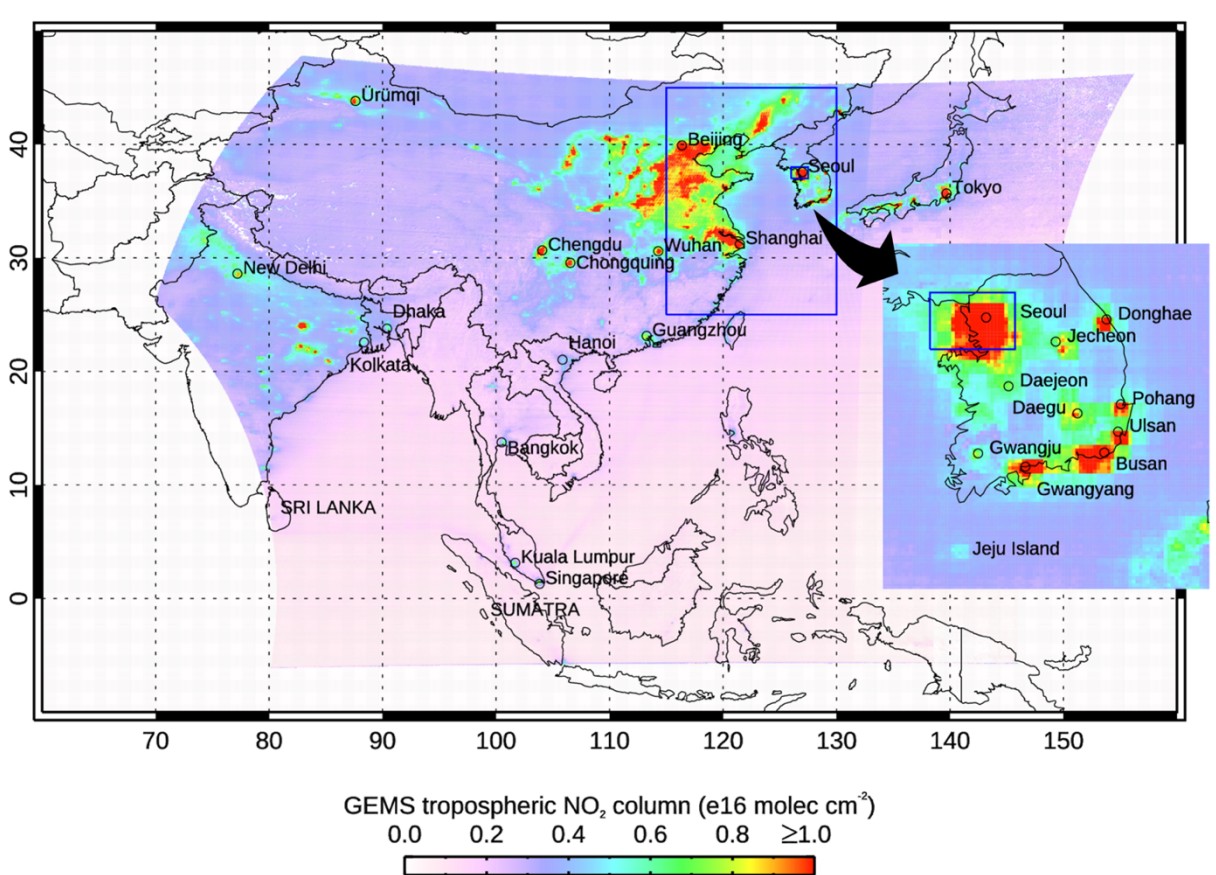



**Figure 1:** Averaged GEMS NO$_2$ TrC for June 2023 showing the full extent of the instrument FOR. Blue boxes indicate the Northeast Asia and Seoul study regions; see text for details. The color discontinuity near 130$^\circ$ E is due to the lower number (2) of daytime hourly observations towards the East compared to the 10 daytime hourly observations in the center of the FOR.

**2.2 TROPOMI**

TROPOMI is a push-broom imaging spectrometer on ESA's Sentinel 5-Precursor satellite in a sun-synchronous orbit with a 13:30 local standard time Equator crossing (Veefkind et al., 2012). TROPOMI achieves close-to-global daily coverage at resolutions down to 3.5 x 5.5 km$^2$, depending on species. NO$_2$, HCHO, carbon monoxide (CO), SO$_2$, O$_3$, methane (CH$_4$), aerosol, and cloud are retrieved from UV-vis and reflected shortwave infrared measurements. We use the TROPOMI operational Level 2 data from Collection 3 that are publicly available through the NASA Earthdata portal (Earthdata, 2024).

**2.3 NASA/ESA PANDONIA Global Network (PGN)**

To capture time resolved measurements of highly variable species such as NO$_2$ in a coordinated manner, NASA initiated a large-scale global monitoring network of (quasi-) autonomous stations with the ground-based remote sensing spectrometer system called Pandora (Herman et al., 2009; Spinei et al., 2018). ESA joined this project in 2018 to form the Pandonia Global Network (PGN), which ensures systematic processing and dissemination of the data in support of AQ monitoring and 125 satellite validation. Pandora instruments measure in the UV-vis and retrieve column amounts of several air pollutants including NO$_2$, HCHO, and O$_3$. Data are publicly available (PGN, 2023).

**2.4 Atmospheric chemistry model framework**

The Multi-Scale Infrastructure for Chemistry and Aerosols (MUSICA) is a new community CTM for simulations of large-scale atmospheric phenomena in a global modeling framework, while still resolving chemistry at emission and exposure relevant scales (Pfister et al., 2020). In this work, we use MUSICA Version 0 (MUSICAv0) which is a configuration of the Community Atmospheric Model with chemistry (CAM-chem) (Tilmes et al., 2019; Emmons et al., 2020) using a spectral element (SE) grid with regional refinement (RR) (CAM-chem-SE-RR), e.g., (Lauritzen et al., 2018; Schwantes et al., 2022). MUSICAv0 is run with a horizontal resolution of 0.0625$^\circ$ (~7 km) over refined regions selected to cover the Korean and wider Asian domain (Jo et al., 2023) and allows near matching of the GEMS pixel resolution of 7 × 7.7 km$^2$ over Seoul (see 135 Fig. S1 of Jo et al., 2023). Chemical processes are all simulated in CAM-chem, which includes chemistry feedback on the meteorology (e.g., aerosol-cloud interactions). To reproduce the dynamics for the January and June 2023 months analyzed, the capability of CAM-chem to nudge the model meteorology to GEOS5 0.25$^\circ$ resolution reanalysis outside of the refined domain is used following Jo et al. (2023). Inside of the refined domain, the wind fields are calculated by the model. We



stress that this work does not aim for exact model simulations of the GEMS data. Rather the model is used to investigate the

processes driving the observed NO₂ diurnal variation.

Date-specific anthropogenic, biomass burning, and biogenic emissions are used in the simulations. The Copernicus Atmospheric Monitoring System version 5.1 (CAMS-GLOB-ANTv5.1) global emission inventory (Soulié et al., 2023) serves as the base anthropogenic emissions inventory along with the NIER/KU-CREATE inventory for East Asia and the Korean peninsula that was produced for the Korea-United States Air Quality Study (KORUS-AQ) field campaign in May-

June 2016 (Jang et al., 2019; Park et al., 2021; Crawford et al., 2021). Global biomass burning emissions are provided as 0.1º resolution daily averages by the Quick-Fire Emissions Dataset (QFED) version 2.5_r1 (Koster et al., 2015) and Fire INventory from NCAR (FINN) v1.5 (Wiedinmyer et al., 2011). Biogenic emissions are simulated using the MEGAN v2.1 algorithm (Guenther et al., 2012), which is incorporated in the Community Land Model (CLM) (Lawrence et al., 2019) and calculated at each model timestep using the model meteorology. Global inventories of anthropogenic emissions are usually

provided as monthly means that are temporally interpolated to a particular day by the model. However, the diurnal variation of emissions also becomes important at the high spatial resolution of the MUSICAv0 refined grid. Diurnal emissions profiles have been derived for different sectors by country in the Emissions Database for Global Atmospheric Research (EDGAR) and can be applied to other inventories (Crippa et al., 2020; Jo et al., 2023).  For our simulations over Seoul, we consider the diurnal variation of emissions using profiles based on area/point and mobile sectors developed for KORUS-AQ and

described in application to modeling over Seoul by Jo et al. (2023). This is shown in Fig. 2 and predicts a pronounced increase in emissions with daytime activity and rush-hour peaks in the mobile emissions.

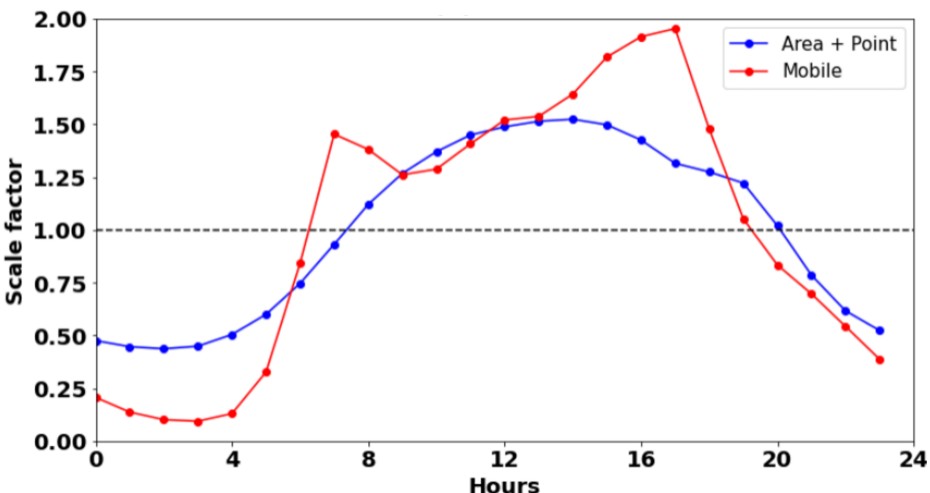

**Figure 2:** KORUS-AQ diurnal emissions profiles for the Area+Point and Mobile sectors.



## 3 GEMS observed NO₂ diurnal variation

### 3.1 Hourly measurements

An example of the ten GEMS daytime retrievals of NO$_2$ TrC over Northeast Asia (115°–130° E, 25°–45° N) for the relatively clear day of 15 June 2023 is shown in Fig. 3. High NO$_2$ is seen over the Beijing region and the industrial areas of the North China Plain. Pollution is also high over Shanghai and the Yangtze Delta with another hotspot over Seoul in South Korea. The most striking first impression of this GEO perspective on atmospheric composition is how large the temporal
variation of the pollution is in magnitude as well as how much the spatial distribution shifts hour-by-hour. In certain locations, changing cloud fields do not permit for all ten possible retrievals to pass the data filters. However, the hourly observations do provide at least some measurements, thus demonstrating an advantage of the GEO perspective. During the winter months when the sun is low in the sky, there are fewer GEMS hourly retrievals. The example is shown in Fig. 4 for 30 January 2023 NO$_2$ TrC over Northeast Asia. Compared to Fig. 3 (which uses the same color scale), the NO$_2$ burden is
considerably higher because of reduced NO$_x$ photochemical loss and NO$_2$ TrC build-up during the day. This results in less diurnal variation compared to the summer months and is discussed in Section 5.

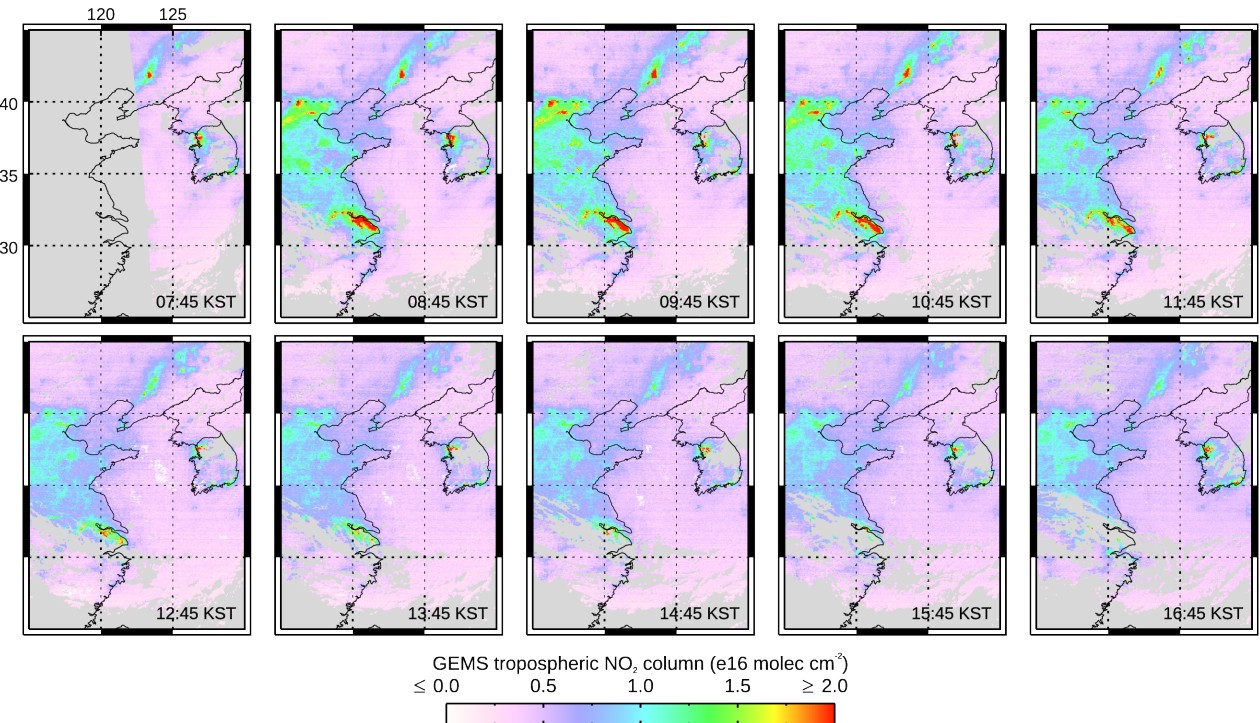

**Figure 3:** Northeast Asia NO$_2$ TrC hourly values, 15 June 2023. Observation times are quoted at Korean Standard Time (KST), i.e., Coordinated Universal Time (UTC) plus 9 hours. Gray indicates no data were taken during night-time or missing
data due to clouds.





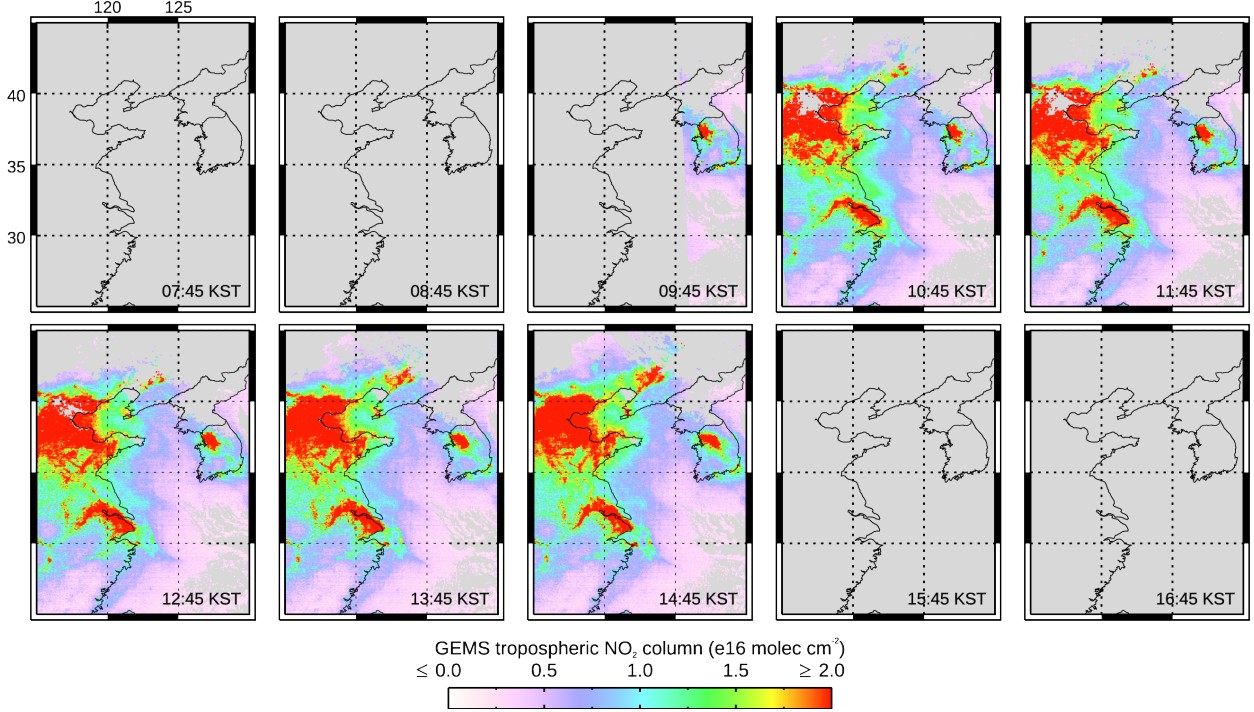

**Figure 4:** Same as Fig. 3 but now for 30 January 2023 NO$_2$ TrC.

## 3.2 Quantifying GEMS NO$_2$ diurnal variation

We have developed quantitative measures of the magnitude of the GEMS NO$_2$ TrC daily absolute and relative variation. The

first of these represents the sum of the absolute change in NO$_2$ TrC over the day at a particular location. Fig. 5 shows the

monthly average of absolute change for June 2023. Because this quantity depends on the number of useful cloud-free

retrievals during the day, it is only calculated for days and locations where there are at least 5 hourly observations, and it thus

represents a lower bound on the total TrC change. No values are mapped east of ~130° E because there are too few daytime

hourly observations at these locations. High variation usually coincides with locations of high TrC, such as industrial regions

and cities, and illustrates the importance that time-resolved observations will have for characterizing changing NO$_x$

emissions and population pollution exposure. Chinese and Korean cities are particularly noticeable as are Indian power

facilities. Ship tracks between Hong Kong and Singapore as well as between Sri Lanka and the northern tip of Sumatra were

previously identified in GOME, SCIAMACHY, OMI, and TROPOMI NO$_2$ data, among others (Beirle et al., 2004; Richter

et al., 2004; Franke et al., 2009; Georgoulias et al., 2020). The ship tracks in Fig. 5 show diurnal variation most likely

because of horizontal dispersion with the increasing afternoon marine boundary layer. Averaging data temporally in this way

at a given location reduces the contribution of transient emission or transport events that can produce significant day-to-day




variability in the NO₂ TrC. The averaged diurnal variation is then primarily dependent on emission and chemistry processes that occur on most days and thus indicates polluted urban regions in particular.

A second measure we consider relates the magnitude of the GEMS NO₂ TrC daily relative variation over multiple hourly observations with respect to the value of the single observation that would be obtained from a LEO instrument such as TROPOMI. For each day of June 2023 and at each location, the absolute deviation of the day's hourly observed NO₂ relative to the observation at 13:45 local time (closest to the TROPOMI overpass time) is calculated and normalized by the observation at 13:45 local time. The daily normalized absolute deviations are then averaged over the month. There are no values in locations such as Japan and India where there is no 13:45 local time observation. This relative NO₂ diurnal variation quantity is shown in Fig. 6 and is seen to be large, often >50% of the 13:45 local time value. The spatial distribution of the relative diurnal variation is similar to the absolute diurnal variation shown in Fig. 5, but in this case, illustrates the percentage diurnal uncertainties in emissions or exposure that might be expected from assuming estimates based on LEO observations. As for the absolute diurnal variation, because this quantity depends on the number of useful cloud-free retrievals during the day it again represents a lower bound on the TrC variation. The apparent high variation over the Pacific Ocean is a result of normalizing by the relatively low NO₂ TrC values in this region.

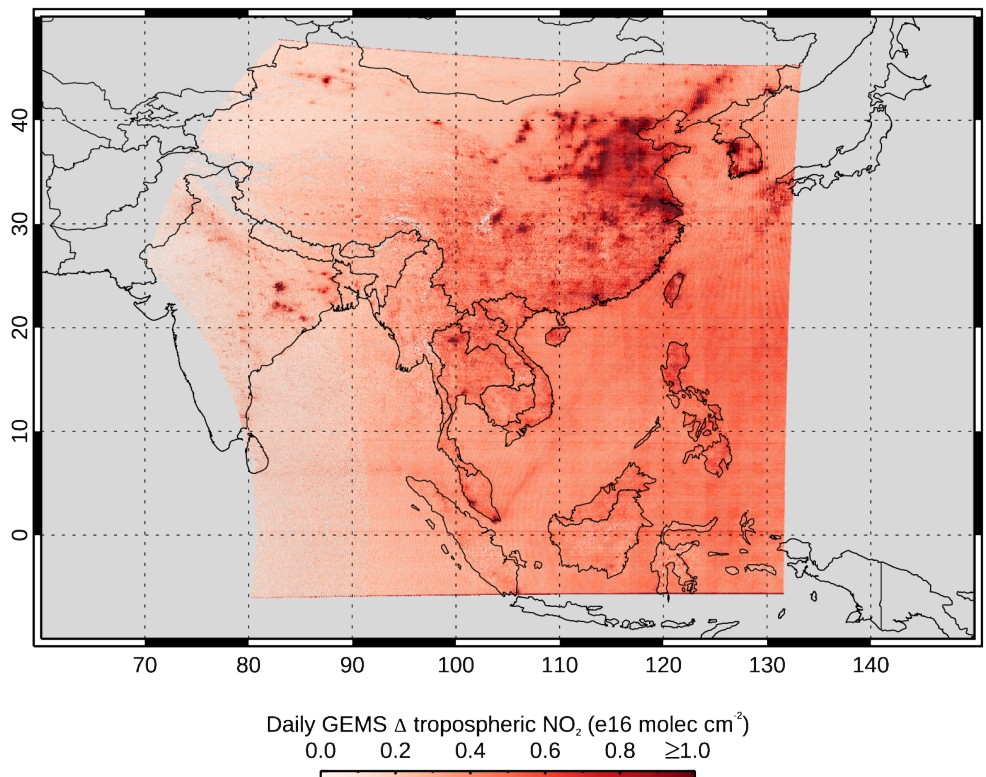

**Figure 5:** Monthly average of the absolute daily change in GEMS NO₂ TrC for June 2023. The region east of ~130° E is not mapped because only data points with 5 or more observations per day were included in this analysis.



The January 2023 monthly average of the GEMS NO$_2$ TrC diurnal absolute daily change and relative variation over the Northeast Asia region are shown in Figs. A1 and A2, respectively. In general, reduced winter photochemistry results in a

smaller diurnal variation in January compared to June. However, over polluted regions, the absolute daily change in NO$_2$ has similar values due to the higher January TrC. A noticeable region of high absolute daily change is also seen in Cambodia which requires further investigation but may be explained by fires or lightning. The Fig. A2 diurnal relative variation map coverage is limited because the 13:45 local time observation is only available between longitudes 113 $^\circ$ E and 132$^\circ$ E.

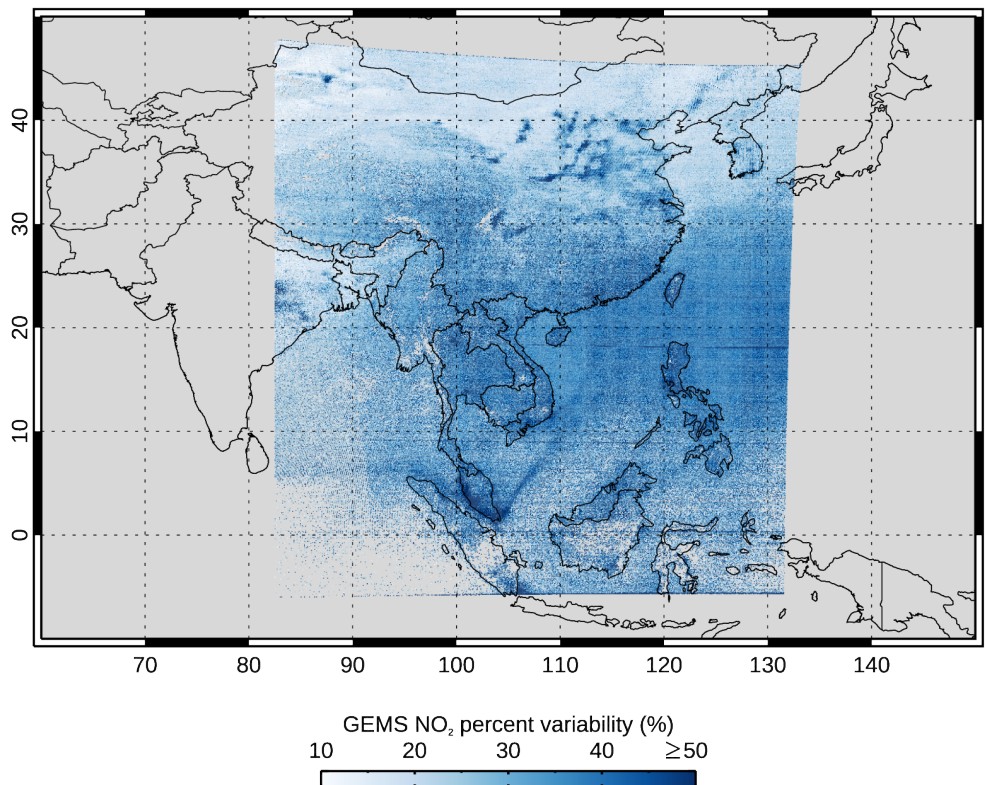

GEMS NO$_2$ percent variability (%)

**Figure 6:** June 2023 monthly average of the GEMS NO$_2$ TrC diurnal variation relative to the 13:45 local time observed

value at each location. Regions west of 90$^\circ$ E and east of ~130$^\circ$ E not mapped because there is no observation at 13:45 local time.

**3.3 GEMS regional and local scale NO$_2$ diurnal variation time series**

We have examined the time series of the GEMS NO$_2$ TrC for various regions and seasons. Fig. 7(a) shows the June 2023 time series spatially averaged over Northeast Asia (see Fig. 1 for regional context). At this time of year, there are a

maximum of ten daylight hourly data points at the center of the GEMS FOR, fewer at the Eastern and Western edges. The number of hourly data points is further reduced by cloud filtering. Day-to-day, the calculated average TrC appears noisy because of the changing amount and nonuniformity of cloud-free coverage, especially when polluted urban areas might be





included in the spatial average one day but not the next. Because of this, the monthly time-averaged diurnal variation (Fig. 7(b)) is the quantity often considered. However, it's important to show the individual daily TrC after filtering for cloudy data

to indicate the information that will usually be available for AQ applications. It should also be noted that cloudy missing points in the daily data tends to 'flatten-out' the apparent diurnal variation shown in the monthly time average and suggests that this quantity be treated with care as it may not capture the full dynamic range of individual day diurnal variation. Despite these considerations, a consistent diurnal cycle is seen on those days with multiple data points. This shows NO₂ TrC decreasing through the morning hours, reaching a minimum in the early afternoon, and then increasing again late in the

afternoon. Little difference is seen between weekdays and weekends at this spatial scale. As discussed in Section 5, we attribute this summer cycle primarily to photochemistry being the dominant driver of diurnal variation since at this regional scale, diurnal variation due to different local-scale emissions and meteorology is averaged-out. This photochemical diurnal cycle is even more apparent when averaging over larger geographical regions although different local times, solar zenith angles and photolysis rates then complicate interpretation. A similar diurnal cycle is also seen in clean regions away from

local sources or transported pollution plumes.

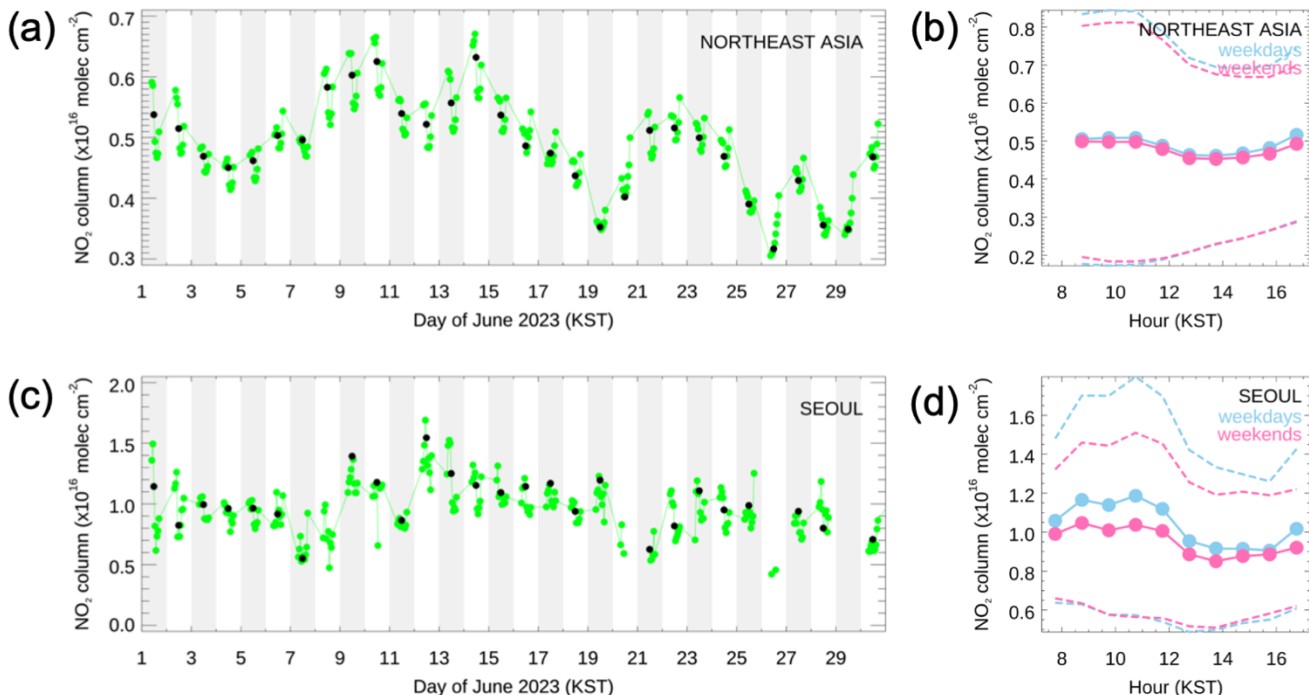

**Figure 7:** June 2023 diurnal NO₂ TrC values spatially averaged over (a) the Northeast Asia region, and (c) Seoul (see Fig. 1 for regional context). Depending on cloud cover, there are up to 10 data points each day and no data at night. The black dot each day indicates the value closet to local noon. The monthly-average weekday and weekend daily NO₂ variation is also

shown for (b) the Northeast Asia region, and (d) Seoul. Note the high standard deviation (dashed lines).



The NO₂ diurnal variation is less consistent at local scale where, in addition to photochemistry, changing emissions and, more importantly, meteorology determine day-to-day variability. Fig. 7(c) shows the time series spatially averaged over Seoul, South Korea (126º–127.5º E, 37º–38º N; see Fig. 1 for regional context). Prior to the launch of GEMS, stagnation events over Seoul had been seen to cause build-up of pollution during the day leading to an afternoon maximum in NO₂ TrC.

This was observed by the Geostationary Trace gas and Aerosol Sensor Optimization (GeoTASO) aircraft spectrometer (Leitch et al., 2014) during the KORUS-AQ field campaign (Judd et al., 2018; Crawford et al., 2021). GEMS observations sometimes show this same diurnal pattern, but as discussed in Section 5, more often show a photochemical 10:00-11:00 morning maximum in NO₂ contributed by urban emissions followed by a decrease and a small late afternoon increase as shown in the monthly time-averaged diurnal variation in Fig. 7(d). The weekend values indicate a similar diurnal cycle to

weekdays with smaller TrC. We note that other work has shown large differences in the NO₂ diurnal variation seen by GEMS between weekdays and weekends for different Asian cities (Park et al., 2022a).

Reduced diurnal variation is shown in the January 2023 GEMS time series spatially averaged over the Northeast Asia and Seoul regions as shown in Fig. 8. As noted above, the apparent large day-to-day differences in TrC results primarily from the varying amount of cloud-free data that enter the spatial average, especially for the largescale Northeast Asia region. Over

both regions, an increase in TrC is usually observed during the day because of reduced winter photochemistry as discussed in Section 5. This is the case for both weekdays and weekends although the weekday TrC values are now clearly greater indicating the persistence of NO₂ resulting from higher weekday NOₓ emissions.

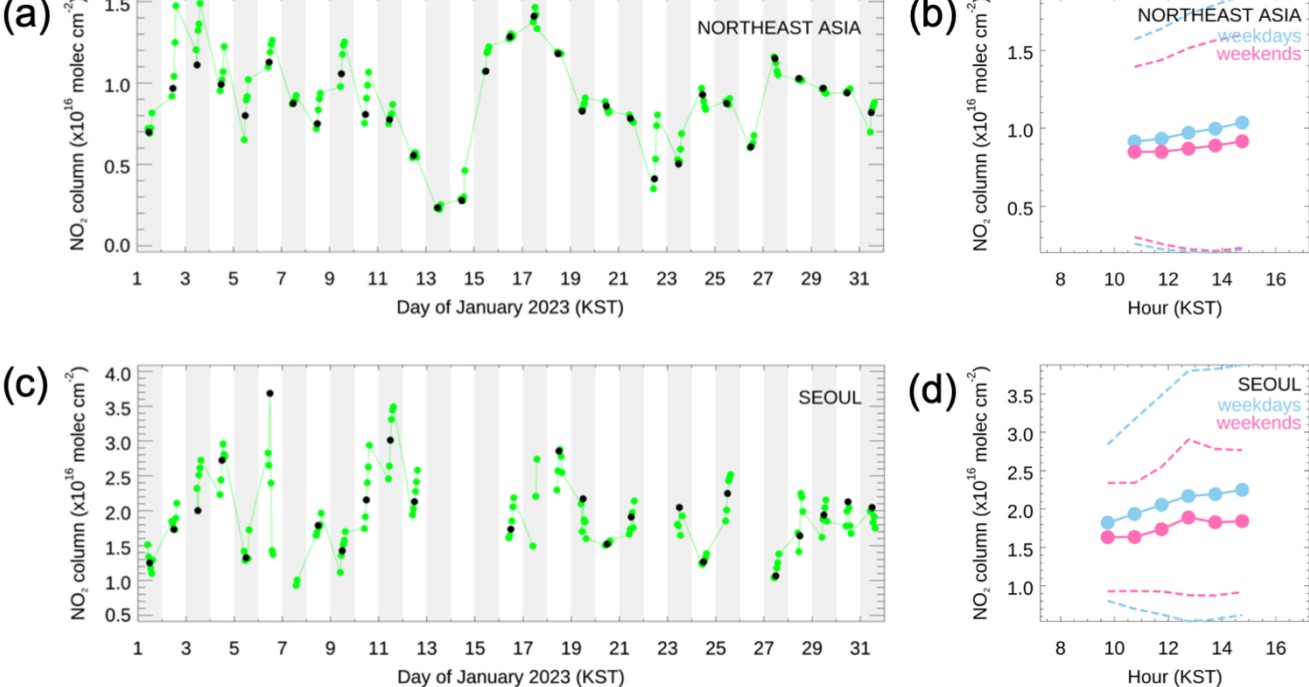



**Figure 8:** Same as Fig. 7 but now for January 2023 diurnal NO$_2$ TrC. Dependent on cloud cover, there are up to 6 data points
each day and no data at night.

## 4 Pandora measurements in Seoul

The Pandora instruments have emerged as the primary source of ground-based measurements for validation of GEMS NO$_2$
and are also used extensively in TROPOMI validation (Kim et al., 2023). The advantage of these sun photometers is that
they provide column retrievals using the same spectral bands as GEMS and have similar measurement vertical sensitivities.
The number of Pandora stations across Asia has been rising rapidly and there are now 34 instruments contributing to the
PGN. Here we use Pandora measurements in Seoul as an independent indication of the diurnal variation.

For our study months, there were two PGN Pandoras making measurements within Seoul, at Seoul National University
(Seoul-SNU, Pandora#149) close to Mt. Gwanak, and about 12 km to the North across the city and closer to the center at
Yonsei University (Seoul-YN, Pandora#54). A comparison of the monthly time-averaged diurnal variation of the NO$_2$ TotC
measurements from GEMS and the Pandora instruments is shown in Fig. 9 for weekdays in June and January 2023. These
also indicate the monthly mean TROPOMI TotC NO$_2$ at the local overpass time. We show Pandora NO$_2$ TotC retrievals that
use direct sun measurements rather than the Multi-Axis Differential Optical Absorption Spectroscopy (MAX-DOAS)
measurements used for the TrC estimate. We also note that our model studies with the Whole Atmosphere Community
Climate Model (WACCM) indicate that over relatively polluted regions such as Seoul, the stratosphere contributes only a
few percent NO$_2$ to the total column. The GEMS and TROPOMI TotC are calculated using a mean of retrievals within 5 km
of the Pandora sites. For the Pandora/satellite measurement comparisons we follow previous work (e.g., Judd et al., 2020;
Lambert et al., 2023).

The GEMS NO$_2$ TotC values are overall larger than their Pandora counterparts with a higher bias in January than June. This
is contrary to what is usually found when comparing pixel-averaged satellite retrievals to local measurements that might
capture small-scale high values (e.g. Herman et al., 2019). Tang et al. (2021) show that this representativeness error can
account for a single Pandora measurement in Seoul being as much as ~25% higher than the GEMS retrieval for the
corresponding pixel. Indeed Kim et al. (2023) found that GEMS V1.0 NO$_2$ TotC measurements tend to be lower than their
Pandora counterparts at less polluted sites south of Seoul. We find that the GEMS V2.0 data positive bias is less than that of
the previous V1.0 data but may still indicate retrieval issues to be addressed in future GEMS data releases. (A preview of the
upcoming GEMS V3.0 NO$_2$ data release with improved AMF calculation and StrC/TrC separation shows closer agreement
with TROPOMI in summer but still an overestimation in winter.) The agreement between Pandora and TROPOMI is good at
both sites and for both months, even though large negative (0 to -50%) biases in TROPOMI NO$_2$ TotC have previously been
reported (Verhoelst et al., 2021). We note here that we find no obvious measurement local time dependence in the
GEMS/Pandora bias for Seoul or for the other Korean sites that we have examined.



Despite the positive GEMS bias, the agreement in pattern of NO₂ TotC diurnal variation captured by the Pandoras and the corresponding GEMS observations is reasonable at both Seoul sites in both months. The column amounts are lower in June than in January and indicate a morning NO₂ maximum followed by a decrease through early afternoon and then a slight increase in the late afternoon. A clear rush-hour peak is not seen in these TotC measurements and is discussed further in Section 5.2. The weekend values (not shown) have a flatter diurnal profile and less variation within each hour, though the

average magnitude is only slightly smaller than weekdays. In January, both sites show a flat or increasing TrC during the day.

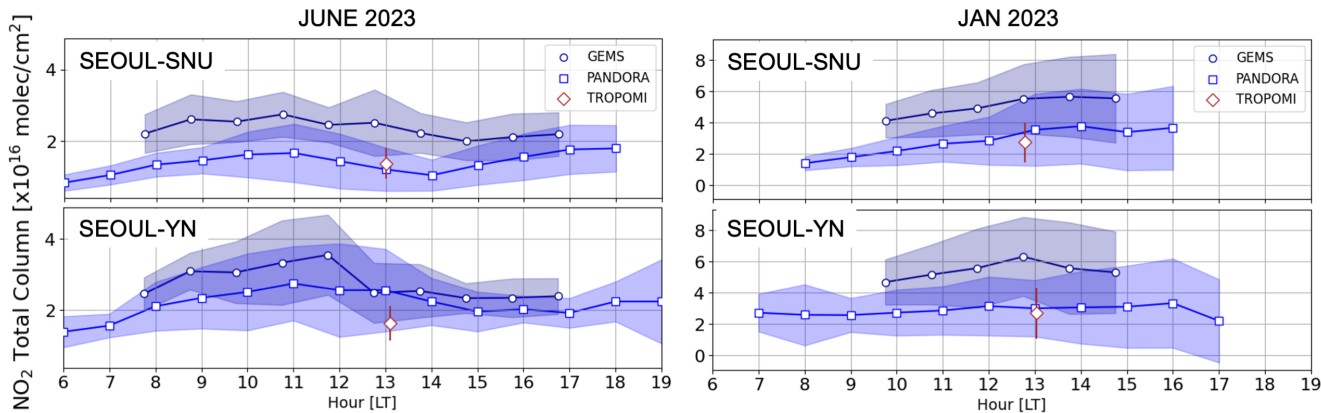

**Figure 9:** GEMS, TROPOMI, and Pandora NO₂ TotC values at Seoul-SNU and Seoul-YN, South Korea, for weekdays in June and January 2023. The GEMS and TROPOMI TotC are calculated using a mean of retrievals within 5 km of the Pandora sites.

The Pandora NO₂ TotC are generally higher at Seoul-YN in June under the prevailing wind from the South at this time of year that blows more pollution from the city center toward this Pandora site. The opposite occurs during January when the prevailing wind is from the Northeast. The same finding for Pandora/TROPOMI comparisons was previously reported by Park et al. (2022b). The GEMS satellite data captures some of this difference between the two sites and suggests more work is needed to assess the capability the of GEMS to resolve pixel-scale urban variability for AQ applications.

## 5 Model studies and discussion

### 5.1 Model NO₂ diurnal variation

We have used the MUSICAv0 model to simulate the NO₂ TrC for the same Northeast Asia and Seoul regions discussed above for June and January of 2023 and the model setup was as described in Section 2.4. For each case, we performed two simulations as a sensitivity test to the assumed anthropogenic diurnal emissions profile. The first assumes constant

anthropogenic emissions during the day (labeled Base), the second uses the KORUS-AQ area/point and mobile sector



diurnal emissions profiles described in Section 2.4 (labeled Diurnal). The average daily NO₂ TrC diurnal variation for weekdays is shown in Fig. 10. The time windows corresponding to the periods during which GEMS retrievals are shown in Figs. 7 and 8 are indicated by the unshaded hours.

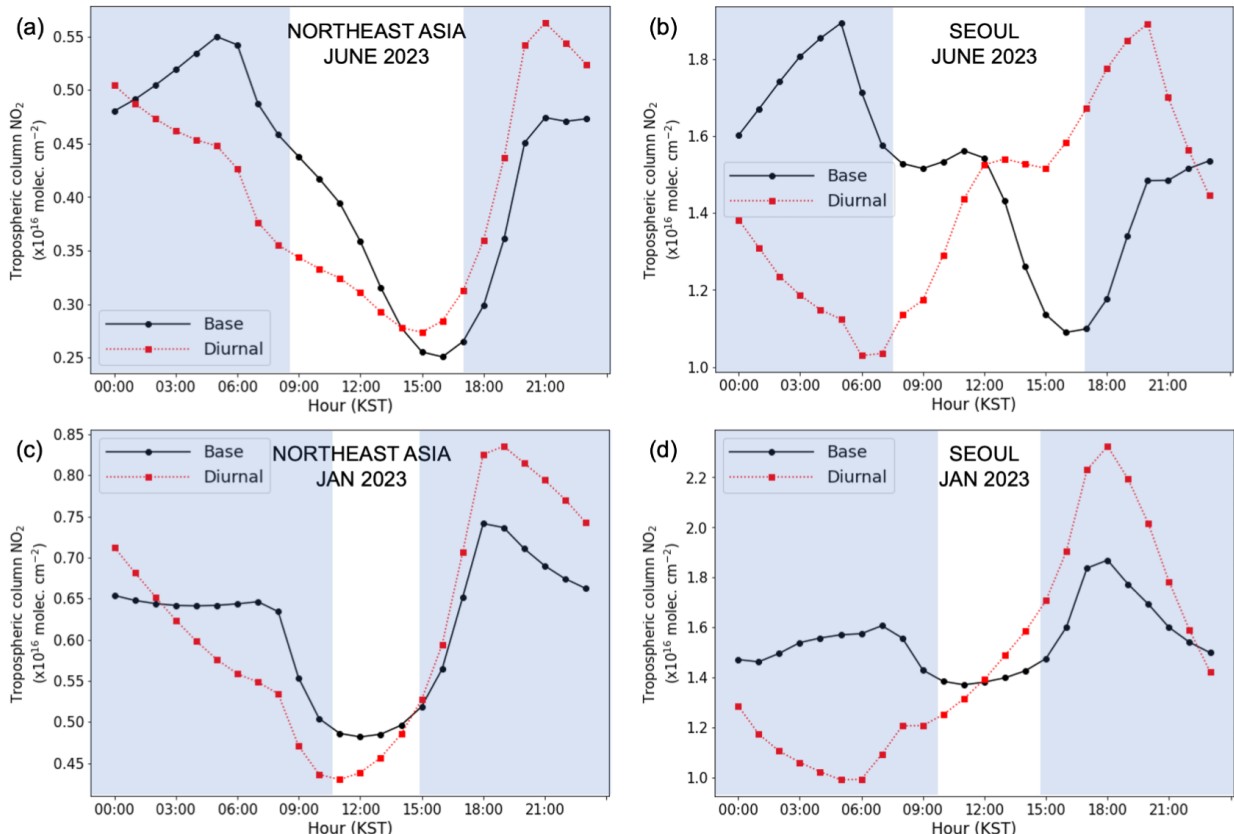

**Figure 10:** MUSICAv0 June 2023 average weekday NO₂ TrC diurnal variation averaged over the Northeast Asia region (left) and Seoul (right) for June 2023 (top row) and January 2023 (bottom row). The time windows corresponding to the periods during which GEMS retrievals are shown in Figs. 7 and 8 are indicated by the unshaded hours.

## 5.2 The role of emissions

We first consider the model results spatially averaged over the Northeast Asia study region and temporally averaged for the months of June and January 2023, Fig 10 (a) and (c), respectively. The model patterns for the NO₂ TrC diurnal variation are in general similar to the corresponding GEMS Figs. 7(b) and 8(b). However, since the model does not include cloud cover and reproduces a consistent daily pattern, the dynamic range of the diurnal variation is greater than the averaged GEMS data. The GEMS data are biased high relative to the model in both months with the difference being largest in January. This may be a GEMS V2.0 retrieval issue as noted above and/or an underestimation of $NO_x$ emissions in MUSICAv0. At this regional scale, the diurnal variation is similar for both the Base and Diurnal simulations despite the very different anthropogenic



diurnal emissions profiles and suggests that photochemistry is the main driver. However, the magnitude of the diurnal variation during the GEMS retrieval window does depend on the emissions profile, ranging from 25-50% in June to 6-17% in January. We note the GEMS $NO_2$ TrC daily relative variation over Northeast Asia for June and January 2023 (Figs. 6 and A2, respectively) falls at the bottom of these ranges suggesting further work is needed to understand the greater model variation. The emissions profile also affects the local hour of the minimum $NO_2$ TrC and is about an hour earlier in the

Diurnal case which matches better with GEMS.

Over the Seoul region, the difference in $NO_2$ diurnal variation between the Base and Diurnal simulations is large, especially in June (Fig. 10(b)). For the Base simulation in June, the shape of the diurnal variation shows a similar photochemical cycle as described above for the Northeast Asia region. A difference is seen in that there is a $NO_2$ morning peak around 10:00-11:00 before the decrease through midday to an afternoon minimum and is like the diurnal variation seen in the

corresponding GEMS data (Fig. 7(d)). This is discussed further in the next Section. In contrast, the Diurnal simulation shows a very different $NO_2$ build-up throughout the day and reflects the shape of the $NO_x$ diurnal emissions profile used in the model, although the pronounced rush-hour peaks seen in Fig. 2 are not evident in the $NO_2$ TrC average even though they do clearly show up in the calculated surface concentration (not shown). However, the fact that the average GEMS $NO_2$ diurnal variation shown in Fig. 7(d) more closely resembles the constant $NO_x$ emissions of the Base simulation suggests that the

hourly changes in the emissions profile may not actually be as large as indicated in Fig. 2. In January, both simulations show increasing TrC during the GEMS retrieval window in agreement with the GEMS data in Fig 8(d). These simulations indicate that the modeled diurnal variation at the city scale will be very dependent on the assumed emissions diurnal profile, accurate characterization of which will be an important part of effectively using the hourly observations from GEO.

### 5.3 The role of chemistry

The underlying photochemical cycle is explained by considering the June 2023 Base simulation TrC $NO_x$ budget analysis shown in Fig. 11(a) for Seoul. After build-up of $NO_2$ during nighttime, photolysis begins with sunrise at 05:00 and the morning decrease is mirrored by the increase in NO with the rise of the $NO_2$ photolysis rate ($j_{NO2}$) curve with increasing solar elevation. Although the $NO_x$ ratio is in photochemical steady state on timescales of minutes, the diurnal change in solar irradiance drives a continual change in the ratio through the day. It should also be remembered that this calculation is for the

TrC relevant to the GEMS retrieval and is therefore representative of a vertically weighted average rather than surface values. After 10:00, NO loss with the build-up of $O_3$ pushes the $NO_x$ ratio toward $NO_2$ at the same time that $NO_x$ is lost to nitrogen reservoirs $HNO_3$, and to a lesser extent, PAN. This balance results in a slight peak in $NO_2$ around 11:00 before a continued decline due to $NO_x$ loss until the minimum at 16:00. Decreasing photolysis results in a subsequent $NO_2$ increase into nighttime with continuing $NO_x$ emissions. The underlying chemistry is similar for the corresponding June Diurnal

simulation (not shown) only in this case, the increasing $NO_x$ emissions during the day result in replacement of most of the atmospheric $NO_x$ that is lost to the nitrogen reservoirs with a consequent gradual buildup of $NO_2$.



In the January 2023 Base simulation shown in Fig. 11(b), atmospheric $NO_x$ shows less variation under conditions of lower photolysis, lower $O_3$, and less daytime conversion to $HNO_3$. The $NO_x$ ratio depends mainly on the change in the $j_{NO2}$ curve under conditions of limited photochemical activity resulting in a 11:00 maximum in NO and minimum in $NO_2$ (as shown in Fig. 10(d)). In the January Diurnal simulation (not shown), $NO_2$ again builds up during the day following the increasing $NO_x$ emissions.

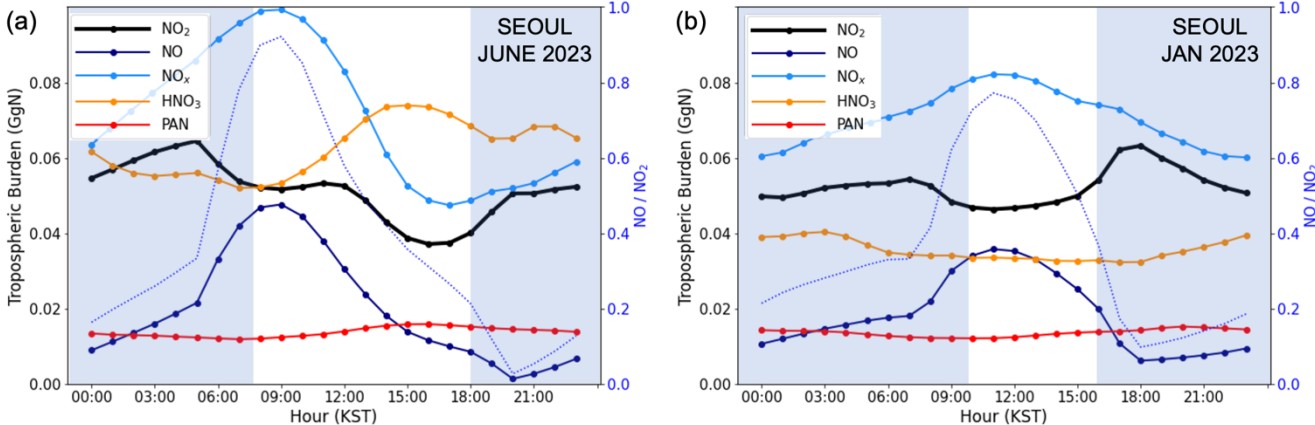

**Figure 11:** MUSICAv0 (a) June and (b) January 2023 average daily $NO_x$ TrC budget analysis over Seoul. Constant anthropogenic emissions during the day (Base simulation). The time windows corresponding to the periods during which GEMS retrievals are shown in Figs. 7 and 8 are indicated by the unshaded hours.

**5.4 The role of meteorology**

The MUSICAv0 average daily $NO_2$ TrC diurnal variation averaged over Seoul (shown in Figs. 10(b) and (d)) results from averaging the diurnal variation during each day of June and January 2023. As noted before, the Base simulation usually shows an afternoon minimum whereas the Diurnal simulation shows a build-up of pollution during the day producing a late afternoon $NO_2$ peak. Both simulations indicate that day-to-day changes in $NO_2$ TrC magnitude depend primarily on meteorology. This is illustrated in Fig. 12 which shows the correlation of the average model $NO_2$ TrC during the day with the model surface layer wind (usually around ~120 m) and also indicates the average wind direction over the city. This is shown separately in plots for June and January calculations for the two years 2022 and 2023 together. For both months, there is a clear anticorrelation (R about -0.7) of $NO_2$ TrC and wind speed. In June, the prevailing wind over Seoul is usually from the South. Shifts to the winds from the West results in lower wind speed and stagnant conditions over the city that permit a build-up of pollution, and this is reflected in the TrC value. Similar anticorrelation is seen in January, although during winter the winds over Seoul are mostly from the Northwest and an occasional change to a weak anticyclonic pattern results in low wind speed and $NO_2$ TrC buildup.





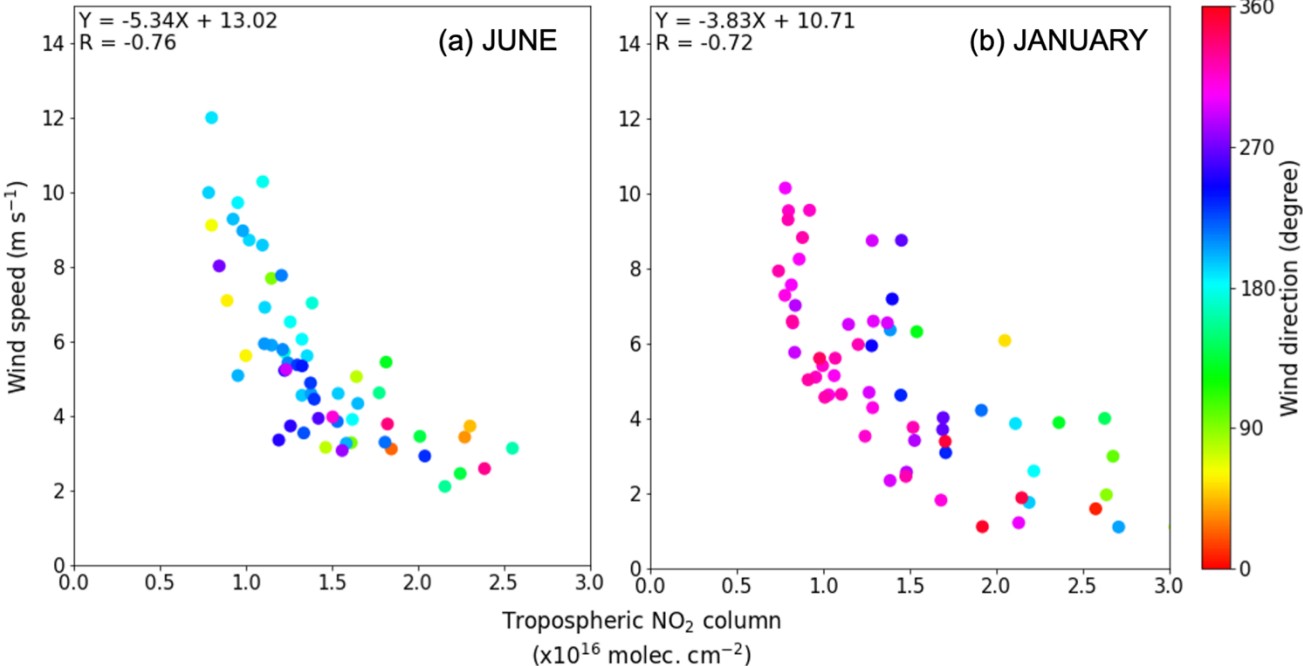

**Figure 12:** Scatterplot of the MUSICAv0 average daily NO$_2$ TrC compared to the magnitude and direction of the prevailing model surface layer wind (usually around ~120 m) over Seoul for June and January combined data for in the two years 2022 and 2023. Constant anthropogenic emissions during the day (Base case).

A similar analysis for the modeled diurnal variation absolute daily change in NO$_2$ TrC discussed in Section 3.2 does not show clear correlation (R about -0.35) with wind speed in either month over Seoul. This suggests that the emission and chemistry processes discussed above are most important in determining the local diurnal variation and do not necessarily require stagnant meteorological conditions. We note that Yang et al (2023b) did see correspondence between low model wind speed and higher winter NO$_2$ diurnal variation, so further investigation would be useful using wind measurements. We have also examined the contribution of incoming transport to the NO$_2$ TrC, although Seoul is unlikely to be a generally representative city in this respect as it is large, fairly isolated, and has very high local NO$_x$ emissions. Upwind GEMS NO$_2$ TrC values are usually about 3–4 times lower than the values retrieved over the city and have relatively flat diurnal variation, suggesting incoming transport contributions to the Seoul NO$_2$ TrC diurnal or day-to-day variations will be small. This might not be the case for longer lived pollutants or when considering concentrations at a specific altitude such as in the free troposphere (Jordan et al., 2020). In cleaner regions downwind of Seoul, the GEMS NO$_2$ TrC do show higher values because of plumes following a high pollution day over the city. Understanding the role of meteorology is going to be important for our next step of relating the satellite retrievals of TrC to surface concentrations. Results from KORUS-AQ showed that



meteorology and PBL dynamics play a large role in determining the extent to which the satellite and ground-based in situ views of pollutant diurnal variation can be reconciled (Crawford et al., 2021).

## 6 Conclusion

Over the last 20 years, LEO observations have provided satellite measurements of pollutants in the atmosphere with
increasing scientific utility, mainly at continental-to-global, weekly-to-seasonal scales. New-generation LEO instruments (e.g., IASI, CrIS, and TROPOMI) have allowed for refinements in both spatial and temporal resolution, to city- and daily-scales. The GEO satellite perspective, with hourly high spatial resolution measurements, represents another major step forward, especially in capabilities for understanding how AQ processes change diurnally at the local scale. The main conclusions of this work are:

1.   GEMS observations show that $NO_2$ TrC diurnal variation can be large (>50% of the TrC) and varies by location, being higher in polluted environments. The $NO_2$ distribution is seen to change hourly and can be quite different from what would be seen in a once-a-day LEO observation. This is demonstrated by the quantitative measures of diurnal variation that we have presented such as the monthly average of the absolute daily change in TrC or the diurnal relative variation in TrC. Along with enabling one or more observations within a day under changing cloud conditions, this demonstrates
410       the advantages of the GEO perspective.

   2.   Regionally averaging GEMS $NO_2$ TrC data emphasizes the diurnal variation due to chemistry since local scale variability due to emissions and meteorology is minimized. Temporally averaging the data hourly emphasizes persistent chemistry and emissions patterns while minimizing meteorological variability.

   3.   In June, $NO_2$ photochemistry is an important driver of diurnal variation, especially at the regional scale. At local scale,
415       $NO_2$ magnitude and diurnal variation patterns change day-to-day, showing the impact of emissions and meteorology. In January, $NO_2$ columns are higher and diurnal variation is lower because of reduced photochemistry.

   4.   Initial comparisons with Pandora measurements over Seoul show a reduction in GEMS V2.0 positive bias with respect to GEMS V1.0 and reasonable agreement on the shape of diurnal variation. The GEMS differences between the two Pandora sites suggests the possibility of resolving pixel-scale urban variation for AQ applications.

5.   Model simulations show high sensitivity to the assumed diurnal emissions profile, especially at local scale. This will have consequences ranging from the assumed $NO_2$ vertical profile used in retrieval AMF calculations to the background model field used for GEMS data assimilation.

   6.   The model indicates an anticorrelation between the surface layer wind speed and the daily mean $NO_2$ TrC which can build up under stagnant conditions.





This work has concentrated on understanding the diurnal variation of the GEMS $NO_2$ TrC retrievals with a CTM. In combination with ground-based remote sensing and in situ measurements, the next step will be to connect the GEO and LEO satellite-derived columns (not only of $NO_2$ but also other trace gas species, particularly $O_3$ and HCHO) to the surface level concentrations. This will allow derivation of top-down diurnal emissions profiles that can be applied to the standard bottom-up emissions inventories. Including this diurnal variation is going to be important for determining true pollutant exposure

levels for AQ studies. The work presented here also provides a path for investigating similar $NO_2$ diurnal cycles in the new TEMPO data over North America, and later over Europe with S-4.

**Appendix A**

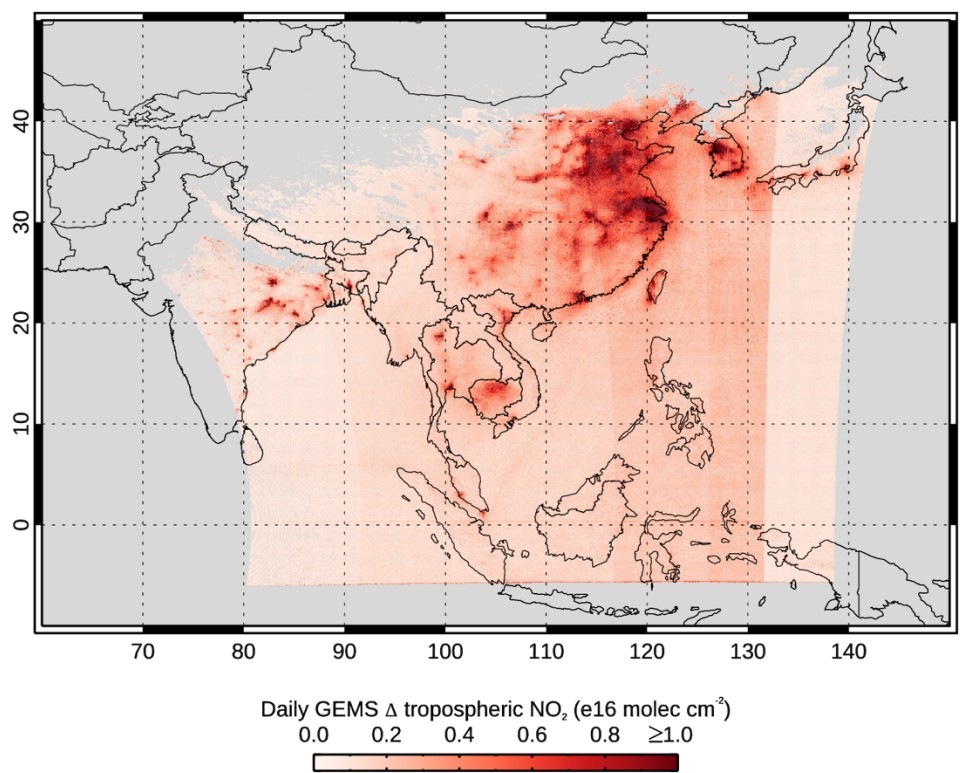

Figure A1: Monthly average of the absolute daily change in GEMS $NO_2$ TrC for January 2023. Data points with 3 or more observations per day were included in this analysis.




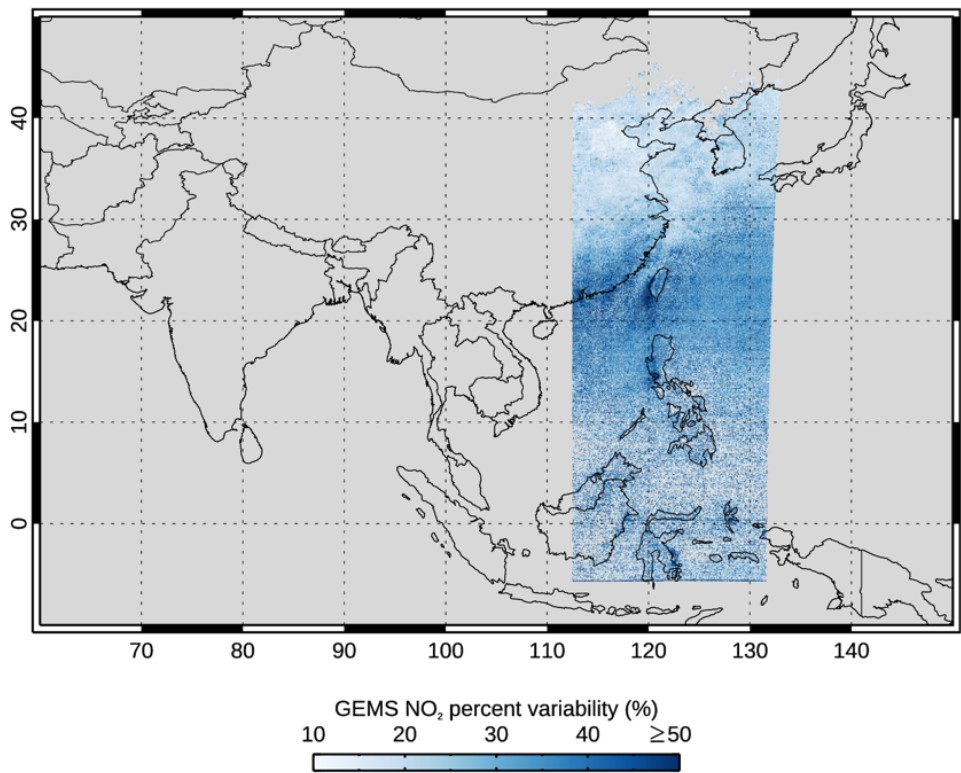

**Figure A2:** January 2023 monthly average of the GEMS $NO_2$ TrC diurnal variation relative to the 13:45 local time observed value at each location. Regions west of about $113^o$ E and east of about $132^o$ E are not mapped due to lacking observations at 13:45 local time.



**Competing interests**

At least one of the (co-)authors is a member of a Copernicus journal editorial board.

**Acknowledgement**

This work was partly supported by Smithsonian Institution subcontract SV3-83021. This material is based upon work supported by the National Center for Atmospheric Research, which is a major facility sponsored by the National Science Foundation under Cooperative Agreement No. 1852977. We would like to acknowledge high-performance computing
support from Cheyenne (doi:10.5065/D6RX99HX) provided by NSF NCAR's Computational and Information Systems Laboratory, sponsored by the National Science Foundation. We thank Doug Kinnison (NCAR/ACOM) for providing WACCM simulations and discussing stratospheric $NO_2$. We thank the TROPOMI and PGN teams for observational data.

**Author contribution**

Conceptualization, DPE; Methodology, DPE, SMA, DSJ, and IO; Formal analysis, DPE, SMA, DSJ, and IO; Data curation,
SMA, DSJ, IO, HL, JP, and HH; Validation, SMA, IO, JK, HL, JP, and HH; Visualization, SMA, DSJ, and IO; Supervision, DPE, HMW, and LKE; Writing – original draft preparation, DPE and SMA; Writing – review & editing, SMA, DSJ, IO, LKE, JJO, HMW, JK, HL, JP, and HH.

**Data availability**

The GEMS L2 $NO_2$ V2.0 data can be obtained on application to NIER
(https://nesc.nier.go.kr/en/html/cntnts/91/static/page.do). The Pandora $NO_2$ data are available from the Pandonia Global Network data archive (http://data.pandonia-global-network.org/). The TROPOMI $NO_2$ data are publicly available from the NASA Earthdata portal (https://search.earthdata.nasa.gov/).

**Code availability**

MUSICAv0 is a configuration of CESM2.2, which is an open-source community model available from:
https://github.com/ESCOMP/CESM. The code modifications for including diurnal cycle of anthropogenic emissions, grid information files, and simulation results are available at [https://doi.org/10.5281/zenodo.8044736].



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
