# Peer review of "Quantifying the diurnal variation of atmospheric NO2 from observations of the Geostationary Environment Monitoring Spectrometer (GEMS)"

_EGUsphere, 2024_

## Author Comment (AC1)

**RC1**: 'Comment on egusphere-2024-570', Anonymous Referee #1, 25 Mar 2024

This study investigates diurnal changes in tropospheric NO2 over northeast Asia and the Seoul metropolitan area, as observed by the geostationary GEMS satellite instrument. The authors attempted to quantify the diurnal changes in GEMS NO2 using two metrics: the cumulative absolute changes in tropospheric NO2 over a day, as well as normalized hourly deviations from local afternoon NO2 values. Diurnal cycles in GEMS tropospheric NO2 for January and June 2023 are compared with ground-based Pandora observations. The authors also conducted model simulations, and discussed the impacts of emissions, chemistry, and meteorology on NO2 over the study area. GEMS, as the first dedicated atmospheric composition instrument in the GEO orbit, is expected to provide valuable insights into the diurnal changes in the emissions and photochemical processes of important air pollutants. This study represents one of the earlier efforts to understand the diurnal changes in GEMS NO2 data, and the results should be of interest to the atmospheric composition and air quality research community. On the other hand, while the paper attempts to combine satellite and ground-based remote sensing measurements with model, the two parts (sections 3-4 and section 5) are only loosely connected. I would recommend major revisions before the paper can be accepted for publication in Atmos. Chem. Phys.

We thank the Referee for their time in reviewing the paper. The comment above about the measurement and modeling parts of the manuscript being loosely connected is addressed along with the Specific comments in the answers below.

**Specific comments:**
As mentioned by the authors, GEMS NO2 retrievals have some known issues and there are improvements that are being implemented. Can the authors use the new version NO2 product for their analysis? At the very least, the authors should demonstrate that with the anticipated changes in the new GEMS product, the major conclusions of the paper would still stand.

We used the latest available Version 2 of the GEMS dataset at the time of writing and this is still the latest available version available to us as of today. We also note that Version 2 is the official data version used by most of the papers in this ACP GEMS special issue excluding independent European and Chinese data processing. Satellite data versions are continually being updated as studies indicate issues and recent correspondence with NIER indicates that they plan release of the next official Version 3 algorithm sometime this year. Future work will assess the conclusions of this paper when new data releases make this possible.

Figure 1: Can the authors comment more on the gradient around 130E? Would the gradient disappear if the authors use the same number of samples across the entire domain? Are the areas east of 130E mainly sampled in the morning? If so, would one expect the mean NO2 for those areas to be higher, as compared with the case with a full daytime sample? In

other words, should one expect that the areas just west of 130E to have smaller NO2 than those to the east, due to sampling differences?

During the month of June, represented in Fig. 1, GEMS retrievals are produced hourly between 7:45 and 16:45 KST (or UTC+9 hours). The region east of the color discontinuity near longitude ~130 is only included in the first 4 of these 10 hourly observations as the FOR moves westward with the Sun. That is, the region east of the color discontinuity is observed only in the morning, as noted by Referee. The color discontinuity would indeed disappear if we discarded 6 of the 10 hourly observations available west of it.

We would expect to see higher NO2 values east of the color discontinuity only if the NO2 amount were homogeneous on both sides of the color discontinuity and it decreased similarly from morning to afternoon in all locations. The lower NO2 values East of the color discontinuity can be explained, for example, by overall higher NO2 amounts to the west that result from more NOx-producing industrial activity, more urban areas, etc.

We expanded the caption of Fig.1 to emphasize the shifting FOR during the day. The caption has also been corrected to indicate that the number of hourly observations to the east of the color discontinuity is 4 (not 2, as stated by the typo in the original caption).

Section 2.4: can the authors comment on the diurnal changes in the GEMS a priori profiles used in retrievals? How do they compare with the MUSICA simulations in this study?

There are diurnal changes in both the GEMS a priori (and MUSICAv0) NO2 profiles as is expected, and the extent to which these changes alias onto the retrieved products is part of the larger concern discussed at Section 2.1 (#104). In particular, the assumed diurnal emissions profile is very important in determining the NO2 profile as discussed in the model sensitivity study in Section 5. The GEMS a priori NO2 profiles come from GEOS-Chem and a note has been added to text in Section 2.1 (#103). As detailed in the new paragraph at the end of Section 2.4 (#171) and in the answer to the Referee's last question below, we would ideally have substituted the MUSICAv0 NO2 as a priori in recalculating the AMF, but unfortunately the required averaging kernels are not available.

Figure 3: there appear to be some horizontal (east-west) stripes in the GEMS tropospheric NO2 over the ocean, can the authors comment on those?

The East-West stripes exist in this data version across the domain similar to the spurious across-track variability issue for OMI. Zhang et al. (2023) comment that this is likely associated with the specific scan modes of GEMS as well as periodically occurring bad pixels. This comment and reference have been added to Section 3.1 (#195).

Line 180: are the absolute changes calculated based on two consecutive hourly observations? If there is a gap between two observations, how is that handled?

We have expanded the discussion of the absolute change in NO2 in Section 3.2 to answer the Referee's question and the equation used for calculation has been added to the text. The absolute daily variation (ADV) of the NO2 TrC change is calculated for adjacent hourly observations, regardless of gaps due to missing data (gaps along a slope would have no effect on the resulting ADV value, although gaps where the slope changes sign may result in an underestimation). This is added to the text (#213).

Figure 5: can the authors comment on the features (hot spots) that can be seen over the oceans (e.g., east of Philippines).

We believe the Referee is referring to the aliasing-like features that are a plotting artifact. GEMS NO2 files from Version 2 do not include the actual corner coordinates of the pixels, just their center coordinate. Because of this, all pixels are mapped as if their size and shape were those of a pixel at nadir. Thus, pixels away from nadir may appear smaller than they should be, producing aliasing-like features in the map.

Lines 252-255 (and Figure 8): some of the day-to-day changes can also be caused by synoptic weather conditions. Do the MUSICA simulations show similar day-to-day changes in NO2?

We agree and this is commented on in Section 3.3 (#281). MUSICA shows day-to day changes in TrC that result from meteorology and this is discussed in Section 5.4. This shows that lower wind speed and stagnant conditions over Seoul permit a build-up of pollution and higher TrC value. The main driver of the apparent day-to-day variability in Figs. 7 and 8, especially over the NE Asia study region is the drop-out of polluted cloudy pixels from the average calculation as discussed in Section 3.3 (#260).

Figures 7 and 8: maybe add Pandora time series to Figures 7c and 8c?

The Pandora plots shown in Fig. 9 are for single Pandora station locations within Seoul whereas the plots in Figs. 7 and 8 are for an average over the Seoul study region (shown in Fig. 1). There is significant variation over the Seoul region as shown by the differences between the two Pandoras at Seoul-SNU and Seoul-YN. The Pandora measurements and the corresponding GEMS data shown in Fig. 9 are also for total NO2 column, whereas the plots in Figs. 7 and 8 are for tropospheric columns as discussed in Section 4.

Section 5: this section appears to be only loose connected to the previous sections and the comparisons between GEMS and model simulations are largely qualitative. Can the authors sample MUSICA simulations using GEMS observation time and cloud filter? Also, can the authors apply the GEMS diurnal change metrics (as discussed in section 3.2) to the model simulations and make comparisons of those metrics between GEMS and MUSICA? Additionally, I'd strongly suggest that the authors use MUSICA simulated NO2 profiles as a priori to estimate air mass factor for GEMS NO2 – this would allow more consistent, quantitative comparisons.

We understand and appreciate the Referee's concerns about the largely qualitative nature of the GEMS/model comparison presented in Section 5. When we started this work, the intention was to perform a quantitative analysis of the GEMS retrievals using MUSICAv0 and to fully use the Version 2 data including the averaging kernels. However, there are known issues with the GEMS V2 processing of the NO2 averaging kernels and the values reported in the operational data files, such that these should not be used as per guidance from NIER. The intention of the GEMS project is to remedy this issue with the next GEMS Version 3 release that will come sometime later this year. Before submission of this paper, we explored the possibility of an alternative processing of the GEMS data to calculate averaging kernels but decided it was beyond the scope of this work and our resources at the time. We then revisited this question over the last month following the Referee's comment and discussed the issues with the wider GEMS team. The conclusion is that the community will wait for Version 3 for reliable averaging kernels and the ability to make quantitative model comparisons using the NIER official product. We decided that waiting for more robust GEMS AK products would inhibit the timely reporting of results that are important to understanding data from the full GEO constellation.

We have added a paragraph at the end of Section 2.4 describing how averaging kernels should ideally be used, the issues with the Version 2 product, and the caveat that the GEMS/modeling analysis in the paper is largely qualitative. We have also added a comment to this effect at the beginning of Section 5. In answer to the Referee's other comments above, without the ability to make a quantitative comparison and to avoid confusion of what we are actually presenting with the model simulations, we think it better to just show pure MUSICAv0 simulations. Filtering the model with the GEMS cloud mask does not make a significant difference due to the large area and monthly averages that we present in Fig. 10 showing the MUSICAv0 diurnal cycle. Similarly, without accounting for the measurement vertical sensitivity through application of the averaging kernel to the model, it's not clear what we learn from applying the diurnal change metrics to the model. The averaging kernels would be needed for the last point about changing NO2 profiles.

We believe that the modeling discussion in Section 5 is still useful even if the comparisons with the GEMS data are largely qualitative. We use the model primarily for sensitivity analysis to capture the pattern of NO2 diurnal variation and the large dependence on assumed diurnal emissions profile and meteorology. The model also provides the 24-hour diurnal chemical cycle context for the variation during the GEMS daylight measurement times. However, if the Referees disagree, we could revise the manuscript removing most of the modeling description (Section 2.4) and results (Section 5) except for the stand-alone chemical analysis presented in Fig. 11, which we would move to the end of Section 3.3.

---

## Author Comment (AC2)

**RC2**: 'Comment on egusphere-2024-570', Anonymous Referee #2, 07 Apr 2024

Edwards et al. present observations of diurnal variations in NO2 from GEMS, one of the geostationary satellites measuring the NO2 column at an hourly time scale. The authors quantify the diurnal variation using two metrics: the sum of absolute changes in the NO2 column over the day and the absolute deviation of the day's hourly observed NO2 relative to the observation at TROPOMI overpass time. The authors also utilize TROPOMI and Pandora measurements to help interpret the NO2 diurnal variation observed in the GEMS observations. The manuscript is well-written but requires some clarification on the context, such as the connections between satellite observations and modeling analysis. I suggest major revisions before publication in ACP.

We thank the Referee for their time in reviewing the paper. The comment above about the clarification on the context of the observations and modeling is addressed along with the other points in the answers below.

1. I suggest adding the equations to describe the matrices used to quantify NO2 diurnal variations. It is very hard to understand what exactly is defined as the "monthly average absolute change" in the figures.

Following the Referee's suggestion, the discussion of the diurnal change in NO2 in Section 3.2 has been significantly expanded and the equations used for calculation have been added to the text.

2. It is worth more discussion on the uncertainty in the GEMS NO2 column observations. What are the uncertainty levels of the hourly, daily, and monthly NO2 column measurements? Is the NO2 diurnal variation described in the paper sensitive to the measurement uncertainty?

The nominal GEMS pre-launch accuracy for the NO2 column is $1 \times 10^{15}$ (molec.cm$^{-2}$) (Kim et al., BAMS 2020). The L2 product contains an error term for the NO2 spectral fit but this does not account for the AMF calculation error and the conversion to NO2 vertical column. Analysis for TROPOMI (van Geffen et al., 2022) identified the latter as the main source of error (around ±25 % over polluted regions). This comment has been added to the description of GEMS in Section 2.1 (#113). However, a positive bias in the GEMS V2 NO2 L2 data is often seen, especially over polluted areas, that has been characterized mainly through Pandora and other ground-based remote sensing comparisons of the type we report in Section 4. We also cite Kim et al. (2023) ACP SI who found a negative bias over clean areas attributable to representativeness error. Lange et al. (2024) (currently in review for the GEMS SI) compares GEMS to ground-based DOAS measurements and finds a median relative difference of +64 % and a correlation coefficient of 0.75. We have added this reference to Section 4 (#322). In short, there are known biases with this GEMS version that are being reported (as here) and that will hopefully be addressed in later data releases. Indeed, the biases were decreased between the Version 1 data and this Version 2

data as we discuss in Section 4. But the pattern of the diurnal variation is usually consistent between GEMS V2 and Pandora, between the different studies in this SI and can be reproduced in shape by the model. This is what we stress in this work - using these first GEO observations to assess the diurnal processes.

3. I can't follow the modeling method section and Section 5. How does the model compare against the GEMS observations? Is the model capturing the diurnal variations observed in the GEMS? If the model is biased, how do you utilize the model to gain a process-level understanding of the observed NO2 diurnal variation?

As discussed in answer to Referee's comment #2, there are known biases with this GEMS version that are being reported but the pattern of the diurnal variation that we investigate is usually consistent between GEMS, other measurements and models. We have added a new paragraph at the end of Section 2.4 (#171) and beginning of Section 5.1 to answer the comment about quantitative comparison between the GEMS data and model Section 5 (#357). This is also discussed in detail in answer to the last question of Referee#1.

4. The introduction section provides a detailed discussion of the satellite's capability of observing the NO2 column, but the result section delves directly into the diurnal variation in the NO2 column. I suggest restructuring the introduction section to highlight the significance of this study.

We have added more description at the beginning of the Introduction in Section 1 (#46) for the motivation for looking at the NO2 diurnal variation and GEMS being the first measurement from GEO to do this.

5. Figure 2: I don't think this figure is necessary for the main manuscript, you can move it to the supplement.

As the Referee questions below in comment #8, we needed to strengthen the connection between the GEMS data analysis and the Section 5 modeling work and discussion. We have now made it clearer in the modeling intro Section 2.4 (#167) that the diurnal emissions profile is used in Section 5.1 to investigate the sensitivity of the model TrC to the assumed emissions. These results are shown in Fig. 10 which are better explained by having the diurnal emissions profile in Fig. 2 as part of the main text. To make this clear, we also added a pointer to Fig. 2 in the presentation of the Base and Diurnal model simulations in Section 5.1 (#355).

6. Figure 9 and section 4: the comparison of the NO2 column between GEMS and other measurements at SEOUL-YN raises concerns about possible bias in GEMS measurement. Why there is a much steeper gradient in the observed NO2 column between 12 and 13 local times from GEMS? Why is it?

We agree that the Figures show a bias. As discussed in Referee's comment #2, there is a known positive bias in the GEMS V2 NO2 L2 data especially over polluted areas that has been characterized mainly through Pandora comparisons of the type we discuss in Section 4. The sharp gradient in GEMS data over Seoul-YN in June between 12:00 and 13:00 is due to the value at 13:00 being anomalously low. This occurs because of the limited number of measurements (2-3 per day) that meet the coincidence criteria with Pandora and then the low number of cloud-free days in the month. A note has been added to Section 4 (#339) to clarify this.

7. Following Figure 9, I wonder if the authors can point out and focus on regions where GEMS can provide more reliable NO2 column observations.

There are regions where the bias between the GEMS and Pandora is much smaller and these tend to be clean areas. Sometimes GEMS even shows a negative bias. But these are also regions where the diurnal cycle is much flatter and of less interest for understanding diurnal processes. We have seen this for our analysis of other Pandora sites, and it is also noted in Lange et al. (2024). We have added a note in Section 4 (#328).

8. Section 5: This part seems barely connected to other sections. More discussion is needed to strengthen the connections between sections.

Please see the answer to the last question of Referee#1 where this point is discussed in detail.